# Translational Models in Glioma Immunotherapy Research

**Alexander L. Ren** [1] , **Janet Y. Wu** [1] , **Si Yeon Lee** [2] **and Michael Lim** [2,*]

1   School of Medicine, Stanford University, Stanford, CA 94305, USA; alexren@stanford.edu (A.L.R.)
2   Department of Neurosurgery, Stanford University Medical Center, Stanford, CA 94304, USA
*   Correspondence: mklim@stanford.edu

**Abstract:** Immunotherapy is a promising therapeutic domain for the treatment of gliomas. However, clinical trials of various immunotherapeutic modalities have not yielded significant improvements in patient survival. Preclinical models for glioma research should faithfully represent clinically observed features regarding glioma behavior, mutational load, tumor interactions with stromal cells, and immunosuppressive mechanisms. In this review, we dive into the common preclinical models used in glioma immunology, discuss their advantages and disadvantages, and highlight examples of their utilization in translational research.

**Keywords:** immunotherapy; glioma; syngeneic murine glioma lines; genetically engineered mouse models; humanized mice; organoids; immune checkpoint inhibitors; dendritic cell vaccines; CAR-T cell therapy; oncolytic virotherapy

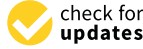



## 1. Introduction

Gliomas are rare cancers of the central nervous system that have an incidence of approximately six per 100,000 people [1]. While gliomas can vary in terms of severity and mortality, the most common primary malignant glioma, glioblastoma, is notoriously deadly with a 5-year survival rate of 6%.

The treatment of glioblastoma has remained largely unchanged over time, and the most recent significant advance was the introduction of the Stupp protocol. This protocol involves surgical resection followed by adjuvant radiation therapy and temozolomide [2]. Despite this, mortality rates for glioblastoma remain high, and thus, much research has been dedicated towards finding new treatment approaches. Immunotherapy is a revolutionary treatment modality for cancer. Inhibitors targeting the T cell immune checkpoint molecules programmed cell death protein 1 (PD-1) and cytotoxic T-lymphocyte associated protein 4 (CTLA-4) have been monumentally impactful for the treatment of various cancers, including metastatic melanoma [3], non-small cell lung cancer [4], and head and neck squamous cell carcinoma [5].

Gliomas display unique immunological properties that differentiate them from other cancer types. They are frequently described as "cold" tumors, implying a lack of tumor-infiltrating lymphocytes. The tumor microenvironment (TME) for intracranial cancers is notably populated by immunosuppressive myeloid cells, with tumor-associated macrophages comprising the plurality of this myeloid population [6]. To improve the efficacy of immunotherapy for gliomas, preclinical research will need to address the full variety of immune cells that occupy the glioma microenvironment, as these cells undoubtedly interact with lymphocytes to reduce their antitumor activity.

Attempts to study gliomas and develop immunotherapeutic treatments require effective modeling strategies to recapitulate glioma biology and reliably predict drug efficacy (Figure 1). Critically, glioma models, and disease modeling more broadly, exist in different levels of complexity. Techniques can range from orthotopic murine transplants with established cell lines to novel three-dimensional organoid culture systems (Table 1). In this

review, we discuss some of the commonly used models to study glioma immunotherapy, assessing some of the benefits and limitations of each model.

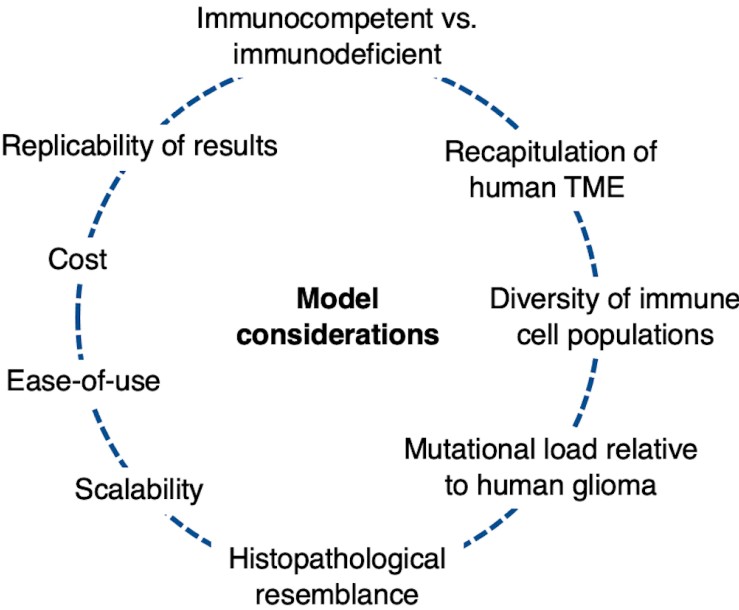

**Figure 1.** Model considerations for glioma immunotherapy research. TME = tumor microenvironment.

**Table 1.** Comparison of glioma models.

| Model | Description | Strengths | Limitations |
|---|---|---|---|
| GL261 | Carcinogen-induced glioma cell line derived from C57BL/6 mice | ○ Widely used, well established model in preclinical glioma research<br>○ Implanted tumors mimic aspects of GBM pathology<br>○ Basal MHC I expression | ○ Immunogenic, more so than human GBM<br>○ Mutational load higher than human GBM<br>○ Becomes more immunogenic with luciferase construct<br>○ No MHC II expression |
| CT-2A | Carcinogen-induced glioma cell line derived from C57BL/6 mice | ○ High GFAP expression corresponding to human astrocytoma<br>○ Less immunogenic than GL261<br>○ Can model CD133⁺ glioma stem cells<br>○ In vivo infiltrating CD8⁺ T cells broadly express exhaustion markers | ○ Less widely-used than GL261<br>○ Becomes more immunogenic with luciferase construct |
| SMA-560 | Spontaneously forming astrocytoma cell line derived from VM/Dk mice | ○ High GFAP expression corresponding to human astrocytoma<br>○ Low basal MHC I expression<br>○ Produces the immunosuppressive cytokine TGF-β<br>○ In vivo infiltrating CD8+ T cells broadly express exhaustion markers | ○ Limited use in the literature<br>○ Mutational load higher than human GBM, though not as much as GL261<br>○ No MHC II expression |

**Table 1.** *Cont.*

| Model | Description | Strengths | Limitations |
|---|---|---|---|
| GEMMs | Mouse models engineered to express mutations in pre-specified genes | ○ Can be used to study pathogenic mechanisms of specific genetic drivers relevant to hypothesis<br>○ Enables the study of spontaneous in situ tumors | ○ TME may not recapitulate the cellular heterogeneity of human tumors, only represents the set of engineered mutations<br>○ Cannot control timing of tumor initiation<br>○ Expensive, difficult to maintain, less scalable than orthotopic models |
| Humanized mouse models | Mouse models depleted of native immune system and reconstituted with human immune progenitor cells | ○ Model human tumor-immune interactions in an animal<br>○ Immune system reconstitution methods have improved | ○ Limited ability to capture the full diversity of human immune cells<br>○ Expensive, difficult to maintain, less scalable than orthotopic models |
| Organoids | 3D cultures derived from primary tissue | ○ Can recapitulate intratumoral heterogeneity of human gliomas<br>○ Potential for high-throughput generation<br>○ Better retains genetic profile of patient tumor used for derivation | ○ Lack vascular endothelial cell and immune cell populations, though methods for introducing these cell types are under investigation |

## 2. Syngeneic Murine Glioma Cell Lines

Cell lines are a popular tool in cancer research, as cancer cell cultures are typically highly robust in their growth and proliferation, and widely used lines are often well-characterized in terms of their genetics and protein expression [7]. Cancer cell lines may be generated from animals or humans, and the particular source species carries downstream implications for the preclinical utility of a cell line.

When murine glioma lines are orthotopically implanted in a syngeneic mouse, one can test immunotherapeutic compounds in vivo with an immunocompetent system. This approach is ideal for studying immunotherapies that rely on enhancing the activity of preexisting immune cells. One caveat is that the mouse immune system and glioma microenvironment differ from their human counterparts, and the translation of preclinical glioma mouse studies remains an ongoing issue in the clinical development of immunotherapeutic agents. Human glioma cell lines can be implanted in immunodeficient mice, but the experimenter loses the ability to study changes in the host immune response towards the cancer. Other advantages of syngeneic glioma lines are their commercial availability and their ability to generate large numbers of implanted mouse models relatively quickly compared to genetically engineered models. In the following sections, we cover some of the more commonly used syngeneic murine glioma cell lines, highlighting their specific application in immunotherapy research. Regarding specific lines under discussion, we will review the murine lines GL261, CT-2A, and SMA-560. These lines were selected for their prominent use in the literature; additional cell lines relevant to glioma research have been covered in other review articles [8–10].

### 2.1. GL261

The earliest models of mouse glioma were achieved via carcinogen induction. Journal articles from as early as 1939 demonstrated that intracranial methylcholanthrene administration could successfully generate brain tumors in mice [11,12]. These methylcholanthrene-induced tumors, referred to as GL261, originally needed to be maintained through serial subcutaneous transplantation to the flank of C57BL/6 mice [13]. By the 1990s, GL261 cells were stabilized as in vitro cell cultures, facilitating future in vitro assays and intracranial transplantation studies [14,15].

GL261 cells closely resemble ependymoblastoma, and this histological appearance was reported to have remained stable over 100 generations [13]. Additional studies since

have noted that GL261 cultures contain undifferentiated cells resembling human GBM cells, and implanted murine GL261 tumors exhibit patterns of angiogenesis, pleomorphism, and pseudopalisading necrosis akin to human GBM pathology [16,17]. In terms of immunological features, GL261 tumors have been shown to be immunogenic, such that injection with irradiated tumor cells prevented tumor formation in 90% of mice when challenged seven days later with normal GL261 cells [15]. GL261 cells exhibit baseline expression of major histocompatibility class I (MHC I), but not major histocompatibility class II (MHC II), which suggests that GL261 is generally susceptible to MHC I-dependent CD8$^+$ T cell-mediated cytotoxicity [15].

As a well-established glioma cell model, GL261 has been used in a variety of preclinical immunotherapy studies. A 1997 study by Plautz et al. used mice bearing syngeneic GL261 intracranial tumors to study the combination of whole-body irradiation and intravenous injection of T cells that had been activated with anti-CD3 antibody or staphylococcal enterotoxin. The authors showed that the mice who cleared their tumors also developed immunological memory to rechallenge with the same tumor cells [18]. GL261-implanted mice were quickly incorporated into dendritic cell (DC) studies for the treatment of gliomas. Preclinical experiments from the early 2000s studied the administration of syngeneic DCs pulsed with GL261 cell lysates and found that such treatment could lead to improved survival in tumor-bearing mice [19,20]. Investigators have also used GL261 to identify adjuvants to DC-based therapies that might enhance their efficacy [8]. Further studies have used GL261 to model the efficacy of chimeric antigen receptor T (CAR-T) cells, a useful approach that circumvents the deficiencies in antigen presentation and co-stimulatory signaling typically seen in high-grade gliomas [21]. Researchers have successfully generated CAR-T cells directed towards GL261 cells engineered to express the tumor-specific epidermal growth factor receptor variant III (EGFRvIII) antigen, with and without cytokine adjuvants [22,23]. GL261 cells were also used to study natural killer cells engineered with a CAR targeting ErbB2 [24]. Finally, GL261 cells have been extensively used in the study of immune checkpoint inhibitor (ICI) therapies. In the last decade, with the emergent success of ICI therapies for other cancer types, investigators frequently test antibodies directed towards checkpoints expressed on T cells. Reardon et al. demonstrated that anti-CTLA-4, anti-PD1, and anti-PD-L1 therapies improved overall survival for luciferase-transduced GL261 tumor-bearing mice. In addition, the combination of CTLA-4 and PD-1 blockade actually yielded a synergistic effect [25]. Additional checkpoint targets that contribute to T cell suppression, such as Tim-3 and IDO1, continue to be identified and queried using GL261 murine models, and ICIs directed towards these targets are furthermore being tested in combination with each other as well as different therapeutic modalities such as radiation [26–28].

Despite the preclinical success of many ICIs in a GL261 model, it is important to recognize that the profound improvements in overall survival largely have not translated to clinical trial outcomes. Anti-PD1 monoclonal antibodies have shown remarkable benefit in GL261 tumor-bearing mice, yet the recent CheckMate trials of anti-PD1 for newly-diagnosed and recurrent GBM patients suggest that there is not a significant clinical benefit relative to or in conjunction with standard-of-care treatment [29–31]. Such results reveal some of the advantages and disadvantages of a syngeneic murine glioma model such as GL261. While GL261 tumors recapitulate some of the major features of human GBM, such as its cellular histopathology, GL261's immunogenicity and TME clearly does not perfectly align with those of human GBM, and it is these areas which may explain the poor clinical translation of ICIs. One study found that luciferase-transduced GL261 (GL261-Luc) cells are more pro-inflammatory and likelier to elicit an antitumor response compared to the non-transduced version of GL261 [32]. Moreover, an analysis of the mutational load generated by GL261 cells suggests that the number of targetable neoantigens is much higher in GL261 compared to most human GBM genetic landscapes [33]. Overall, these factors highlight that GL261 is more immunogenic than human GBM, which would increase the false positive rate of preclinical studies and hamper their predictive capacity for clinical success.

*2.2. CT-2A*

Similar to GL261, CT-2A was generated via intracranial injection of the carcinogen 20-methylcholanthrene in a C57BL/6J mouse [34]. The line was originally maintained by serial subcutaneous transplantation over many generations and bore histological features akin to malignant anaplastic astrocytomas [35]. Immunohistochemical analysis revealed strong GFAP expression among CT-2A cells which corresponds to the high GFAP expression seen in human astrocytomas [36]. Immune phenotyping of CT-2A tumors found low enrichment of genes related to immune response pathways compared to GL261 tumors, reinforcing prior observations that the CT-2A line is much less immunogenic relative to GL261 [37]. CT-2A neurospheres were also specifically shown to express the brain tumor stem cell marker CD133 and the stem cell markers Oct4, Nanog, and Nestin [38]. The expression of these stem-like markers suggest that CT-2A can be a useful cell line for testing therapeutics that target brain tumor stem cells, which may contribute to treatment resistance in high-grade glioma [39].

While less widely used in preclinical research compared to GL261, CT-2A has been used to study a variety of immunotherapeutic modalities. CT-2A has been frequently employed in oncolytic virotherapy studies, suggesting that this line may be more susceptible to viral infection. Barnard et al. constructed an oncolytic herpes simplex virus-1 (oHSV-1) vector expressing a DC-targeting, fms-like tyrosine kinase 3 ligand (Flt3L). Intratumoral injection of the oHSV-1 in CT-2A glioma-bearing C57BL/6 mice improved survival, and the efficacy was thought to stem from the activation of DC precursors and generation of a pro-inflammatory effect [40]. Another study examined the use of the Semliki Forest virus-4 (SFV-4) as a vector for oncolytic virotherapy, which provided the unique advantage of being semi-resistant to type 1 interferons. The investigators inserted microRNA target sequences into the SFV-4 (SFV4miRT) to reduce the virus's ability to infect normal neurons but preserve its oncolytic function. They demonstrated that CT-2A glioma-bearing mice survived longer and experienced delayed tumor growth after a single intravenous injection of SFV4miRT [41]. However, the benefit of this vector may be due in part to the weaker antiviral response and reduced type I interferon levels observed in CT-2A [42].

With regards to immune checkpoint blockade, CT-2A has been frequently used to study the response to PD-1 blockade. Multiple studies identify reduced efficacy of anti-PD1 treatment in CT-2A glioma-bearing mice compared to GL261, again highlighting the reduced immunogenicity of CT-2A [43,44]. Notably, when Liu et al. profiled the immune cells within the tumor microenvironment of mice bearing CT-2A and GL261, they found that the percent of CD8[+] T cells expressing the checkpoint molecules PD-1 and Lag-3 were significantly higher in CT-2A relative to GL261 [44]. Khalsa et al. separately found that more than 70% of CD8[+] T cells in CT-2A tumors were doubly positive for Lag-3 and Tim-3 [37]. This abundant expression of CD8 checkpoint markers may partially explain the muted response of CT-2A to anti-PD1 monotherapy. The relative resistance to anti-PD1 monotherapy makes CT-2A a useful model for studying combination therapies that attempt to enhance the efficacy of checkpoint blockade. For example, the combination of 4-1BB (a T cell co-stimulatory molecule) agonism and PD-1 blockade decreased exhaustion and increased the effector function of tumor-infiltrating lymphocytes in CT-2A tumors, leading to vastly improved survival compared to each treatment in isolation [45]. In a recent study, Khan et al. noted that CT-2A tumors have few functional infiltrating CD4[+] T cells, which likely drives the profound exhaustion of infiltrating CD8[+] T cells observed with CT-2A. They further found that addition of a CD40 agonist could overcome the CD4[+] T cell dysfunction in CT-2A tumors and greatly increase survival when combined with anti-PD1 therapy [46].

As previously discussed, the addition of a luciferase construct can increase the immunogenicity of the GL261 cell line [32]. Similarly, CT-2A-Luc glioma-bearing mice were found to have a higher number of infiltrating T cells in the brain compared with CT-2A controls. Additionally, CT-2A-Luc mice treated with anti-PD1 survived much longer than untreated CT-2A-Luc mice, whereas CT-2A mice treated with anti-PD1 experienced no

survival benefit [47]. This finding should encourage caution when interpreting the positive results of preclinical immunotherapy studies that employ luciferase-expressing cell lines.

*2.3. SMA-560*

Unlike the carcinogen-induced approach used to initially generate GL261 cells, SMA-560 cells are distinct glioma line derived from a spontaneously-forming astrocytoma in a VM/Dk mouse [48]. The line was stabilized as an in vitro tumorigenic cell line in 1980, whereas it was previously maintained via serial in vivo intracerebral transplantations [49]. Histologically, SMA-560 tumors display high cellularity and an invasive border and can be stained for GFAP, confirming the cells' astrocytic lineage. SMA-560 cells exhibit basal, albeit low, expression of MHC I but not MHC II. Uniquely, SMA-560 was shown to produce transforming growth factor beta (TGF-β), an immunosuppressive cytokine that is expressed in human gliomas and is thought to decrease the proliferation and activation of cytotoxic T cells [50]. Woroniecka et al. further showed that tumor-infiltrating lymphocytes in SMA-560 murine tumors exhibited upregulation of immune exhaustion markers, most notably PD-1, Tim-3, and Lag-3. Moreover, their analysis found that almost half of tumor-infiltrating CD8$^+$ T cells in SMA-560 were triply positive for the three aforementioned checkpoints, and the pattern of co-expression was similar to that observed in human GBM samples [51]. SMA-560 cells were also shown to carry a relatively high mutational load compared to human GBM, though not quite as high as GL261 [33]. Overall, the constellation of these immunological findings suggests that SMA-560 can be a reliable glioma model for assessing experimental immunotherapies.

Tran et al. demonstrated that administration of a TGF-β inhibitor improved survival of an orthotopic SMA-560 mouse model. TGF-β inhibition reduced the invasive capabilities of glioma cells, increased T cell infiltration into the tumor, and increased CD107a surface expression on CD8$^+$ T cells in the cervical lymph nodes, indicating that TGF-β has broad-ranging effects on both the cancer and immune cells in the SMA-560 murine model [52]. Miller et al. transfected SMA-560 cells with a plasmid encoding soluble CD70, which acts as a ligand for the co-stimulatory molecule CD27 on CD8$^+$ T cells. They showed that soluble CD70 prolonged the survival of VM/Dk mice bearing these genetically-modified cells [53]. Like GL261, SMA-560 has been used in early preclinical research on dendritic cell therapies. Vaccination using DCs pulsed with SMA-560 homogenate improved the survival of SMA-560 tumor-bearing VM/Dk mice due to an enhanced cytotoxic and antibody-mediated immune response [54]. In terms of CAR-T research, CAR-T cells directed towards EGFRvIII demonstrated profound curative potential in SMA-560 orthotopically-implanted mice [55]. Additional data from this study found that cured mice also displayed an immune response towards EGFRvIII-negative tumors, suggesting that infused CAR-T cells could stimulate broader immunity to a variety of tumor-associated antigens [55]. A study by Przystal et al. demonstrated a unique approach of combining a CSF1R inhibitor and an anti-PD1 drug in SMA-560 mice. CSF1 is a cytokine that is important for the survival of tumor-associated macrophages, and the inhibition of its cognate receptor is intended to target the myeloid cell population in the glioma microenvironment [56]. Przystal et al. found that anti-CSF1R antibody could potentiate the efficacy of anti-PD1 therapy, suggesting that targeting both the myeloid and lymphoid compartments could improve symptom-free survival in SMA-560-implanted mice [57].

Overall, GL261, CT-2A, and SMA-560 have been established as central murine glioma models for immunotherapy research. While they have been thoroughly characterized, there are potential drawbacks with their use, including the variable immunogenicity between lines and higher mutational load relative to human GBM. These qualities should highlight the need for more rigorous preclinical validation of candidate targets for ICIs, potentially using these murine lines in conjunction with some of the other model systems discussed in this review.

## 3. Genetically Engineered Mouse Models

Rather than diving into individual variants of genetically engineered mouse models (GEMMs), which have been discussed in depth in other reviews [58–61], we will instead address GEMMs as a whole and discuss overarching themes regarding their applicability to immunotherapy research.

We first discuss the principles of generating GEMMs and the information that can be gleaned from their usage. GEMMs are engineered to overexpress oncogenes or decrease expression of tumor-suppressor genes, and thus, they represent a targeted approach to spontaneous tumor initiation. Because tumor generation is predicated on specific genetic manipulations, GEMMs can be a useful model for studying basic tumor biology and evolution. GEMMs can be generated via germline modification or somatic cell gene transfer [62]. Germline modification involves the induction of gain-of-function or loss-of-function mutations in zygotes or embryonic stem cells, with subsequent breeding strategies to maintain transgenic mouse lines [59,62,63]. While initial modification methods involved a pronuclear injection of a DNA construct into the zygote or injection of modified embryonic stem cells into the mouse blastocyst, newer methods have applied CRISPR-Cas9 gene-editing technology to more precisely generate double strand breaks in the blastocyst stage [64,65]. Further manipulation of tumorigenesis can be achieved through the use of inducible recombinase systems (such as the CreER-LoxP system) that can provide spatial and temporal control over the expression of inserted transgenes [66]. These inducible models also prevent the issue of embryonic lethality if the gene under investigation is critical to the early development and/or viability of the mouse. Somatic cell gene transfer refers to the use of viral vectors to transfect specific cell populations with genetic material [62]. One of the more widely used systems for somatic cell gene transfer in brain tumor research is the RCAS/tv-a system, which relies on an avian retrovirus (RCAS) that targets cells which have been engineered to express the cognate receptor (tv-a) [66,67]. A seminal paper by Holmen and Williams applied this RCAS/tv-a system to demonstrate the role of Ras signaling in Kras-induced glioblastomas, illustrating how GEMMs can be useful for studying the genetic alterations and downstream molecular signaling cascades that contribute to tumorigenesis [68].

GEMMs have been used to study the basic immunology of brain tumors, since the spontaneous in situ nature of these tumors likely enables more faithful recapitulation of the human tumor microenvironment [69]. Kong et al. used the RCAS/Ntv-a transgenic mouse system to induce glioma formation via co-expression of PDGF-B and Bcl-2. The investigators found that intratumoral phosphorylated-STAT3 expression and macrophage influx correlated with prognosis in their GEMM model, akin to clinical observations of patient outcomes [70]. They also tested an immunotherapeutic agent—the p-STAT3 inhibitor WP1066—and demonstrated survival benefit with an associated reduction in macrophage infiltration [70]. Alghamri et al. recently used a GEMM expressing mutant IDH1, ATRX, and TP53 to profile tumor-associated myeloid cells. They identified an expansion of granulocytes with increased G-CSF expression, the inhibition of which resulted in restored immunosuppression. Thus, the authors use a GEMM to characterize the myeloid cell milieu of gliomas bearing IDH1 mutations, which can be leveraged for the development of targeted immunotherapies [71]. Zamler et al. generated a Qk/Trp53/Pten triple-knockout (QPP) GEMM of human GBM via an inducible Cre-Lox system. This triple-knockout combination was previously shown to preserve the stemness of glioma stem cells and promote glioma invasion and migration [72]. In addition to exhibiting human GBM histopathology features—including pseudopalisading necrosis, microvascular proliferation, and positive S100-β and GFAP staining—this QPP GEMM also recapitulated the immune cell populations in human GBM, which consists of predominantly immunosuppressive myeloid cells with smaller lymphocyte populations [72,73].

GEMMs provide unique advantages for translational modeling purposes. Many GEMMs can capture the histological and immunological features of certain human gliomas, mimic their invasive quality in the brain, and reveal phenotypic insights attributed to

specific genetic alterations [74,75]. Since GEMMs can be engineered for spontaneous tumorigenesis, one can avoid the confounding effects that accompany orthotopic injection of syngeneic tumor cells, such as artificial disruption of the blood-brain barrier [9]. In addition, our ability to introduce targeted genetic modifications makes GEMMs a useful tool for studying rarer glioma types that are otherwise difficult to model with orthotopic transplants, such as diffuse intrinsic pontine glioma [76]. However, there are also particular disadvantages to the use of GEMMs. Since GEMM tumors typically reflect a homogenous group of tumor cells bearing specific mutations, they fail to capture the intratumoral heterogeneity observed in human gliomas [77]. This caveat reduces the clinical relevance of genetically engineered models, which limits the translatability of GEMM preclinical results. Compared to orthotopic mouse model studies, GEMM-based studies can often be more resource-intensive and require a longer timeframe for tumor growth, and there is also less control over the timing of tumor initiation [9,61,78].

## 4. Humanized Mouse Models

Once again, a wide range of methods exist for the generation of humanized mouse models [79], but this review will focus on broader themes and the application of these models to brain tumor immunotherapy research. As noted earlier, the transplantation of human cells into mouse models requires the mice to be rendered immunodeficient, otherwise the murine immune system will reject the foreign tissue. Thus, experiments involving patient-derived xenografts (PDX) require immunodeficient mice, which would not be amenable to studying immunotherapeutic mechanisms. Humanized mice offer a means of modeling human immune responses towards human cancer cells in an in vivo context. Humanization of mice refers to the reconstitution of the mouse immune system with human immune cells via engraftment of human tissue, human peripheral blood mononuclear cells (PBMCs), or CD34$^+$ human hematopoietic stem and progenitor cells (HSCs) [66,80,81]. Depletion of the native mouse immune system was originally achieved primarily through irradiation or the use of nude mice, which have impaired thymic development and are thus deficient in T cell development [80]. Immunodeficient mouse strains such as NSG-SGM3 have become more widely used for humanization and have shown to be more effective hosts for the stable engraftment of human CD34$^+$ cells and the development of diverse human immune cell populations [82,83].

Clearly, the appeal of humanized mouse models for immunotherapy research stems from the ability to study human tumor-immune interactions in vivo. This can be incredibly powerful for preclinical studies in translational research, as such mice can be used to demonstrate the safety and efficacy of agents in a clinically relevant setting [84]. Humanized mice have been widely used for mechanistic studies as well as translational purposes. Zhai et al. used NSG-SGM3-BLT humanized mice that were intracranially implanted with patient GBM xenografts to study the intratumoral expression of IDO1, an enzyme that is implicated in cancer-related immunosuppression. They identified tumor-infiltrating CD4$^+$ and CD8$^+$ lymphocytes in their PDX humanized mouse model and further observed that T cell depletion led to reduced IDO1 expression, suggesting that T cells may increase IDO1 expression in human GBM [85]. With regards to immune checkpoint inhibitors, Ashizawa et al. transplanted human PBMCs into an MHC gene-double knockout NOG humanized mouse model (NOG-dKO) and used this model to assess the efficacy of anti-PD1 monoclonal antibody treatment. The investigators subcutaneously implanted the human glioblastoma cell line U87 and observed a trend suggestive of reduced tumor growth following administration of anti-PD1 antibody [86]. There are multiple caveats when interpreting the results of this study, including the choice to use the U87 cell line, which the authors acknowledge exhibits low expression of phosphorylated STAT3 and PD-L1, and the decision to subcutaneously rather than intracranially implant the cancer cells. The location of glioma implantation is crucial, as subcutaneously grown gliomas can elicit stronger antitumor immune responses compared to intracranial gliomas [87]. Klawitter et al. recently tested the combination of oncolytic virotherapy and checkpoint

blockade in a humanized mouse model. They intracranially implanted immunodeficient NSG mice with one of two human GBM cell lines and subsequently transplanted human PBMCs. Intratumoral injection of the experimental oncolytic adenovirus XVir-N-31 induced immune-related tumor cell death and increased lymphocyte infiltration, which was further enhanced with the addition of anti-PD1 treatment [88]. Notably, their humanized mouse model inevitably developed graft versus host disease, and therefore, they could not conduct survival studies, highlighting a disadvantage of this model system for preclinical studies.

There are additional limitations with the use of humanized mice. In the case of CD34+ HSC engraftment, the immune system reconstitution is frequently incomplete as murine cytokines and growth factors cannot adequately promote the development of human immune cell lineages [89]. While transgenic mice, such as NSG-SGM3, have been engineered to express some human cytokines, this solution incompletely recapitulates the full range of cytokines needed to mimic the immune cell populations of patients [90]. Furthermore, human T cells are educated in the mouse thymus and become MHC restricted, which may impair human HLA-related antigen presentation and T cell activation [81,89]. Some solutions attempt to address this problem, such as employing the bone-liver-thymus (BLT) model of humanization in which fetal liver and thymus tissues are co-implanted alongside CD34+ HSCs or using transgenic mice that express human HLA alleles. However, these solutions frequently present their own issues, including limited availability of patient tissue (in the case of the BLT model), mixed results in achieving hematopoietic reconstitution, and significant time and resource investment in optimizing such models [89–91].

## 5. Organoids

Organoids represent an exciting new direction in disease modeling that may complement the aforementioned glioma models. The term "organoid" refers to the technique of culturing pluripotent stem cells derived from human tissue within a 3D-matrix [92]. These cells can be guided in their development through the addition of mitogens, nutrients, and small molecule modulators to recapitulate the differentiated cell types and organization of the tissue of origin [92,93]. Lancaster et al. created a 3D culture system for generating cerebral organoids which successfully formed heterogenous brain regions and captured some of the cortical developmental patterns of humans [93]. Cerebral organoid culturing systems have since been adapted to model glioma development. Hubert et al. initially isolated CD133+ glioma stem cells from patient GBM samples and generated organoids from these cells. The GBM organoids exhibited spatial heterogeneity including a rapidly-dividing periphery and a hypoxic core region, similar to the pathology observed in human GBM [94]. After the organoids were cultured for several months, the investigators dissociated the organoids to single cells and orthotopically implanted them in the brains of immunodeficient NSG mice. Notably, organoid-derived tumors displayed a diffuse and infiltrative phenotype, unlike tumorsphere-derived tumors that displayed a solid, uniform growth pattern [94]. Subsequent methods of glioma organoid modeling have included genetic engineering of normal brain organoids to induce spontaneous tumorigenesis and the introduction of tumor cells into normal brain organoids via co-culture, which can be particularly useful for modeling the human tumor microenvironment in vitro [95–97].

Glioma organoids are a powerful model due to their potential for robust, high-throughput generation, adaptability to both in vitro and in vivo studies, ability to represent the organization of the tumor microenvironment, retention of the parental tumor genetic profile, and recapitulation of human GBM-like features including intratumoral heterogeneity and an invasive phenotype [98–100]. There are, however, several limitations with this model. While brain organoids can achieve recapitulation of regional structures and cellular arrangement, mature organoids still primarily resemble fetal brain tissue [99]. Patient-derived GBM organoids may be difficult to reliably culture and propagate depending on the quality of tumor tissue used and the composition of the sample [98]. Glioma organoids lack several important compartments such as vasculature and immune cell populations [100,101]. Ongoing research efforts are working to optimize co-culture pro-

tocols of tumor organoids and human PBMCs to populate organoids with immune cells. Neal et al. generated air–liquid interface organoids using patient-derived tumor tissues encompassing over 20 different tumor types. Their method used mechanically dissociated tissue fragments to preserve both tumor and stromal cells, including infiltrating T cells and tumor-associated macrophages. The authors successfully used this platform to model immune checkpoint blockade in vitro by treating both human and mouse tumor organoids with anti-PD1 antibody, which stimulated activation, expansion, and cytotoxicity of tumor-infiltrating lymphocytes [102]. As glioma organoids are a relatively nascent technology, there have been few attempts to study glioma immunotherapy with this tool. A critical paper by Jacob et al. sought to use GBM organoids to model responses to a variety of tumor treatments, one of which being CAR-T cell immunotherapy. They generated organoids from microdissected fresh GBM tumors without single-cell dissociation, thereby preserving the original tumor architecture [98]. GBM organoids exhibited EGFRvIII expression and were co-cultured with EGFRvIII-targeting CAR-T cells to test their efficacy against a solid tumor model. The investigators observed CAR-T cell expansion and activation along with EGFRvIII+ tumor cell death in co-cultures, indicating that GBM organoids may be an effective in vitro platform for testing CAR-T and other immunotherapies going forward [98]. As the use of glioma organoids becomes more widespread and the technology evolves to better integrate clinically relevant immune responses, this model will likely become an important tool for preclinical immunotherapy research.

## 6. Conclusions

A variety of models exist to study tumor-immune cell interactions and test the efficacy of immunotherapies directed towards gliomas (Table 1). Syngeneic orthotopic mouse models remain a mainstay of glioma immunotherapy research, primarily because of their thorough characterization, widespread use, and scalability. GEMMs are a useful model for assessing the role of specific mutations in gliomagenesis and can be appropriate for addressing targeted mechanistic questions. Humanized mouse models afford the ability to study the human immune antitumor response in an animal model; however, they can be challenging to establish and maintain and are an imperfect representation of the human immune cell repertoire. Organoids are a novel strategy for closely recapitulating human glioma features in an in vitro setting. While still premature in its preclinical applications, extensive research effort is being devoted to further improving organoid models. No single model is perfect, and rigorous preclinical studies will likely incorporate multiple models to validate positive hits. The decision of which models to use will depend on the immunological mechanism in question and the suitability of the model to provide a translationally meaningful readout.

**Author Contributions:** Conceptualization, M.L., A.L.R. and J.Y.W.; investigation, A.L.R.; writing—original draft preparation, A.L.R.; writing—review and editing, A.L.R., J.Y.W., S.Y.L. and M.L.; supervision, M.L. All authors have read and agreed to the published version of the manuscript.

**Funding:** This research received no external funding.

**Conflicts of Interest:** M.L. reports research support from Arbor, BMS, Accuray, Biohaven, and Urogen. M.L. is a consultant for VBI, InCephalo Therapeutics, Merck, Pyramid Bio, Insightec, Biohaven, Sanianoia, Hemispherian, Novocure, Noxxon, InCando, Century Therapeutics, CraniUs, MediFlix, XSense, as well as a non-research consultant for Stryker. M.L. is a shareholder of Egret Therapeutics and participates on the data safety monitoring board of Cellularity. M.L. holds patents related to the combination of focused radiation and checkpoint inhibitors, the combination of local chemotherapy and checkpoint inhibitors, and checkpoints for neuro-inflammation. The authors have no other conflicts of interest to declare.

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
