# Peer review of "Translational Models in Glioma Immunotherapy Research"

_curroncol, doi:10.3390/curroncol30060428_

Round 1
Reviewer 1 Report
I would like to suggest the authors including another section for the patient derived xenograft of glioma models and their progress and applications on the drug development for glioma therapy.
The English quality looks good to me.
Author Response
Point 1: I would like to suggest the authors including another section for the patient derived xenograft of glioma models and their progress and applications on the drug development for glioma therapy.
Response 1: We thank the reviewer for their suggestion on adding content on patient-derived xenografts. We have expanded our humanized mice section to highlight how this model presents a solution for studying immune responses to patient-derived xenografts or human glioma cell lines, which would otherwise be transplanted into immunodeficient mice.
Reviewer 2 Report
The authors present an update on “Translational Models in Glioma Immunotherapy Research”.
Immunotherapy has shown encouraging results in selected cancers, while the uniqueness of cerebral environment slowed its application in the treatment of brain tumors. Then, translational models are essential but reproducing the complex interactions “human brain-tumor” is challenging, furthermore deepening the immune interfaces.
The authors reassume these issues and pros and cons of the various available models, murine and organoid, are clearly illustrated. However, to facilitate an easier comparison among the existing translational opportunities, the data reported in Table I should be represented more schematically: I suggest subdividing the “Notable Characteristics” column into three-four vertical columns in which are summarized, for instance, 1) technical model description 2) strengths 3) limits, so it would be easier and rapid a comparison among the different models.
Overall, the paper is well organized and useful for researchers.
Author Response
Point 1: However, to facilitate an easier comparison among the existing translational opportunities, the data reported in Table I should be represented more schematically: I suggest subdividing the “Notable Characteristics” column into three-four vertical columns in which are summarized, for instance, 1) technical model description 2) strengths 3) limits, so it would be easier and rapid a comparison among the different models.
Response 1: We thank the reviewer for their insightful comments. We have revised the table to make it easier to read, per the reviewer’s suggestions.
Reviewer 3 Report
Sound review of models, however, except for organoids, models appear to have major problems. As the authors note, little progress has been made in improving glioma survival time. Perhaps new models are needed or research focus needs to be on analysis of human glioma patients with longer survival time.
Author Response
Point 1: Sound review of models, however, except for organoids, models appear to have major problems. As the authors note, little progress has been made in improving glioma survival time. Perhaps new models are needed or research focus needs to be on analysis of human glioma patients with longer survival time.
Response 1: We thank the reviewer for their comments. We hope the reader is satisfied with the current state of the manuscript.
Round 2
Reviewer 1 Report
The authors have addressed my comments. The manuscript is ready for publication.